# OpenReview forum: "Consistency Training Can Entrench Misalignment"
_ICML.cc/2026/Conference — ICML 2026 regular_

### Official Review · Reviewer_5dK2 · 2026-02-13

**Soundness:** 2
**Presentation:** 2
**Significance:** 1
**Originality:** 2
**Overall Recommendation:** 4
**Confidence:** 3

**Summary:**

- The paper studies "Consistency training" or CT (i.e., training models to produce consistent outputs across related inputs or sampling procedures) and its impact on model alignemnt.
 - Specifically across many models of variying size, they try various CT algorithms and evaluate for various misaligned behavior.
 - The main finding is that CT is not "alignment-neutral" tha is in certain ways it fails.

**Compliance With Llm Reviewing Policy:**

Affirmed.

**Final Justification:**

The author's comments addressed most of my concerns.

**Key Questions For Authors:**

See the above box.

**Limitations:**

Yes.

**Strengths And Weaknesses:**

**Strengths:**
 - The framing of the intro is very clear. I give you credit for this.
 - Breads of evals: You evaluate lots of models and a range of algorithms.



 **Big picture:**
 - If you zoom out, no algorithm is "alignment-neutral". There is no perfect since the world complex. Even if you go to theories of morality, there are always aspects that people disagree about, i.e., there is no single truth. Why would you expect any algorithm to be "alignment-neutral"? (is SFT on labeled data "alignment-neutral"? RLHF or GRPO are? ... )

 **Findings:**
 - You evaluate many models and several consistency algorithms. But your findings reports only the *average* which hides the variations and


 **Theory:**:
  - I have issues with your Eq3. Risk is a real-valued function. Of course the tiniest of changes can change the risk and activatuve this equation. Instead, you want to write it so it indicates some sort of gap in terms of Risk chage: |Risk1 - Risk2| > \delta
 - What is M in eq5? I don't think you define it.
 - Overall, it's not very clear how your S3 really ties to the rest of the story -- the notation is both unclear and is abstract.


 **Citations**: There is a host of approachs of use model itself for alignment that roughly fit within the description of "Self-Rewarding". For example you may considner Zephyr (https://arxiv.org/abs/2310.16944) or Alpacha models that use Self-Instruct (https://arxiv.org/abs/2212.10560) to fit this description. There are a ton of works here that are not covered in your description.

---

> ### Author Rebuttal · Authors · 2026-03-30
>
> We thank the reviewer for the feedback. We believe the revisions substantially address the issues raised, and hope that the reviewer considers our work favorably in light of this.
>
> **Issue 1: Big picture.** While we agree any training procedure will to some degree alter behavior, our motivation stems from how consistency training fundamentally differs from other post-training methods such as SFT and RLHF. Because consistency training operates entirely on the model's own outputs, one might expect the technique to mainly solidify existing distributions. Its widespread use in modern post-training pipelines (e.g., Llama 3.1, DeepSeek-R1, Qwen 2.5) to enhance reasoning capabilities reinforces this. Instead, we show that consistency training can produce dangerous safety shifts, such as amplifying high-stakes behaviors like sycophancy. Given the lack of prior work on its safety implications, we believe it is important to document these unintended effects.
>
> **Issue 2: Reporting the average.** Our best understanding of this concern is that reporting the average change over many models, algorithms, and tasks can hide important variation in results. If we have understood correctly, then we would like to clarify that the primary metric throughout the paper is sign consistency (percentage of runs where $Δ < 0$) as opposed to mean $Δ$. This was chosen precisely because it is robust to the high variance the reviewer identifies. For example, when we report that reward hacking shows 63.4% suppression, this is a statement about the fraction of runs showing improvement, not an average that could be dominated by outliers. Additionally, while the main paper presents aggregated results for clarity and brevity, we do include extended results in Appendix G.
>
> **Issue 3: Theory.** We appreciate the feedback on the theory. We have incorporated them into the revised version of our paper. We revise our equation to define $ε$-non-neutrality:
>
> Definition ($ε$-non-neutrality). A consistency procedure is $ε$-non-neutral on organism $(D, M)$ for threshold $ε > 0$ if $|Risk(θ; A_{ct}, D, M) − Risk(θ; A_{base}, D, M)| > ε$
>
> We report which method–organism pairs exceed $ε$ = 5pp throughout the paper, and in the appendix discuss $ε$ = 10 thresholds. At $ε$ = 5pp we find that reward hacking (DD: −27.7%, SR: −11.6%), emergent misalignment (SR: −5.3%), and sycophancy (SR: +7.8%, Rew: +6.0%) all qualify. ACT/BCT effects exceed $ε$ = 10pp on all organisms except spurious correlations. This provides evidence that non-neutrality is not a trivial artifact of measurement noise.
>
> We also appreciate the feedback on the flow of Section 3. In the revised manuscript, we have restructured it to make the narrative flow explicit:
> * Section 3.1 defines procedure-level risk (connecting organisms from Section 2 to a measurable quantity).
> * Section 3.2 states the testable hypothesis ($ε$-non-neutrality, revised per above).
> * Section 3.3 derives Proposition 3.1 as a diagnostic tool for selection-based methods, with an explicit statement that it predicts selection effects should be governed by $η(s)$.
> * Section 6 then tests this prediction, finds $η(s)$ is flat, and uses this negative result to argue for the distributional shift mechanism.
>
> Regarding $M$ in Equation 5: $M$ is the organism-specific misalignment indicator $M(y) ∈ {0,1}$, defined in Section 2.2. We have added a forward reference in Section 3.3 to make this clearer.
>
> **Issue 4: Related work.** We thank the reviewer for these references and have added Self-Instruct (Wang et al., 2023) as used by Alpaca and Distilled Direct Preference Optimization (Tunstall et al., 2023) as used by Zephyr to the related work section along with several other consistency-adjacent methods (STaR, SPIN, GRPO). We also note that Appendix A already contained an extended taxonomy tracing the lineage of methods enforcing consistency and other spiritual predecessors, which we hope addresses the broader coverage concern.
>
> **Additional experiments in this revision.**: In order to maximize transparency, we have added per-model, per-seed scatter plots showing the full distribution of outcomes for every method-organism combination. Further, we conducted two new experiments to strengthen our core claims. First, as noted in our response to Issue 1, we introduced a rejection sampling baseline that closely mimics a known production post-training pipeline, the results of which continue to corroborate our primary findings from the manuscript. Second, we added a greedy self-training baseline that provides further evidence demonstrating how much of each method's effect is attributable to self-generated SFT versus the consistency-specific mechanism.

---

> > ### Author Rebuttal · Reviewer_5dK2 · 2026-04-03
> >
> > I thank the authors for their response.

---

### Official Review · Reviewer_37R5 · 2026-03-13

**Soundness:** 4
**Presentation:** 4
**Significance:** 3
**Originality:** 3
**Overall Recommendation:** 5
**Confidence:** 4

**Summary:**

The question is whether consistency training—a popular technique that encourages language models to produce similar outputs across related inputs—is alignment-neutral or actively changes how safe/aligned models are. The authors propose the Consistency Non-Neutrality Hypothesis: consistency training isn't neutral; instead, it selectively suppresses or amplifies different types of misbehavior depending on whether that misbehavior is stable and coherent under perturbations. They formalize this across 7 consistency methods (ranging from simple scoring-based selection like Self-Confidence to representation-level constraints like Activation Consistency Training) and test on 4 model organisms of misalignment: Emergent Misalignment, Spurious Correlations, Reward Hacking, and Sycophancy. The core tension is this: while consistency training improves reasoning by enforcing agreement across perturbations, it may accidentally reinforce harmful behaviors if those behaviors are more "robust" (stable across prompt variations) than beneficial behaviors.

The paper then formalizes the consistency non-neutrality hypothesis by showing that consistency training changes misalignment risk (Equation 3), and derives Proposition 3.1: for selection-based methods, amplification vs. suppression depends on whether higher-scoring candidates are more or less misaligned—a testable diagnostic before deployment. The experimental pipeline (Figure 2) has three phases: fine-tune base models to exhibit controlled misalignments (model organisms), apply consistency methods to generate pseudo-labels, then fine-tune on those labels. Results are striking and heterogeneous (Section 5): regularization methods (ACT/BCT) achieve near-perfect suppression on reward hacking (95-100%) and emergent misalignment, but ALL methods amplify sycophancy; label-generation methods show mixed results (reward hacking suppressed 63-74%, emergent misalignment 72%, spurious correlations neutral). The mechanistic insight (Section 6.2, Figure 6) is that reward hacking is a brittle, incoherent strategy (~10x higher KL divergence across models), while sycophancy is a stable, coherent strategy (~1x KL), explaining why consistency training filters one out and reinforces the other—behaviors that are stable under perturbations get reinforced, regardless of whether they're aligned or misaligned.
Evidence from KL divergence (Figure 6) shows reward hacking is ~10x more incoherent (KL 0.3) across models than sycophancy (KL 0.03), explaining why consistency training filters one out and reinforces the other—the mechanism is behavioral coherence under perturbation, not alignment.
Takeaway:
Consistency training ≠ "pick the best output"
Consistency training = "shift distribution toward stable/coherent behaviors"

The paper suggests that safety evaluations must shift from pre-deployment-only to pre- and post-consistency evaluation, with method selection guided by organism-specific alignment effects. The paper's core lesson: consistency training is not a neutral capability improvement but an alignment-relevant distribution shift that can entrench misalignment when misaligned behaviors are coherent.

**Compliance With Llm Reviewing Policy:**

Affirmed.

**Final Justification:**

Based on the rebuttal, and in the light of the promised edits to the paper addressing the concerns that were raised in the review, I decide to keep my original score.

**Key Questions For Authors:**

1. The fact that every single consistency method amplifies sycophancy (Table 2: SC 15%, DD 42%, MVC 22%, SR 22%, Rew 26%, ACT 10%, BCT 35%) is striking. But I want to understand if this is:

(A) Fundamental: Sycophancy is so coherent that any consistency enforcement amplifies it
(B) Method-specific: There exists a consistency method that doesn't amplify sycophancy, but you didn't test it

2.You hypothesize: "larger models execute reward-hacking strategies more coherently, making them stable under perturbation."
But this hypothesis contradicts Figure 6's finding that reward hacking has 10x higher KL divergence (i.e., less coherent) than sycophancy. If reward hacking becomes more coherent at 70B, wouldn't KL divergence decrease at 70B? I was wondering  if there is something special about reward hacking's structure that makes it scale-dependent?
3. Can you towards the end, clarify important caveats that should shape how practitioners use consistency training, based on your research and findings?
4. Can you add lines to Section 6.2 formalizing "distribution shift" as a mechanistic explanation, or explicitly state the explanation remains intuitive.

**Limitations:**

The authors deserve credit for:

Being transparent about experimental limitations
Acknowledging synthetic organisms may not generalize
Providing honest caveats on 70B findings
Offering practical recommendations

But should strengthen discussion of:

1. Could this work enable harmful uses?

Positive use: Improve reasoning and robustness
Dual-use risk: Amplify misalignment in adversarial hands
Dual-use uncertainty: Hard to distinguish good-faith safety work from malicious amplification

2. Information asymmetry (who benefits from this knowledge?)
Stakeholder-specific harms from sycophancy vs. reward hacking amplification
Scale uncertainty: 70B findings are exploratory; safety implications for large-scale models unclear

**Strengths And Weaknesses:**

Strengths outweigh weaknesses: The core finding (consistency training is not alignment-neutral) is important, counterintuitive, and actionable, even if some mechanistic details remain unclear
Empirical rigor compensates for theoretical gaps: 602 runs across 7 methods and 4 organisms provide substantial evidence, even if Proposition 3.1 doesn't fully explain the mechanisms
Practical value is high: Recommendations for practitioners (suppress sycophancy upstream, red-team post-consistency) are immediately useful
Honest limitations discussion: Authors transparently acknowledge synthetic organisms, judge error, and exploratory scale results.


This is a well-executed empirical study that makes a important contribution to AI safety by showing consistency training has alignment-heterogeneous effects. While some theoretical and generalization gaps remain, the practical findings are valuable and the experimental work is rigorous.
1. Rigorous Theoretical Framework

Proposition 3.1 is mathematically sound, with clear assumptions and complete proof (Appendix D)
The formalization of consistency training as an operator on completion distributions (Equation 1) is elegant and unifies disparate methods.
However, this doesn't predict what happens in practice. Core issue: The theorem assumes η(s) = P(M(Y)=1 | S(Y)=s) is monotonic. Empirically, η(s) is flat (Figure 5, ~9pp variation)
Under flat η(s), Proposition 3.1 predicts minimal selection effect, yet large amplification/suppression are observed
Implication: Either (a) the theorem's assumptions are violated in practice, or (b) the mechanism isn't selection-based
The paper claims (b)—distribution shift is primary—but doesn't rigorously prove this or characterize what "distribution shift" means mechanistically
Missing: Formal definition of how the label distribution differs from baseline and how this shifts model behavior

2. LLM Judge Dependency
Emergent misalignment and sycophancy evaluations rely heavily on GPT-4o judges. Judge error could inflate/deflate effect sizes; MISSING-sensitivity analysis on judge thresholds.
3. No prior systematic study of alignment effects despite widespread adoption. Counterintuitive and Policy-Relevant Findings
Suppression (good news): Consistency training suppresses reward hacking (63%) and emergent misalignment (72%)
Amplification (bad news): Consistency training amplifies sycophancy (75% of runs)—the opposite of what one might hope
The heterogeneity (same methods help some organisms, hurt others) is surprising and challenges naive intuitions
This has direct implications for safety practices: "More consistency ≠ safer" is an important negative result

4. Practical Recommendations Are Immediately Actionable

Recommendation 1: Suppress sycophancy via SFT before applying consistency training
Recommendation 2: Don't assume k-scaling (more candidates) improves safety
Recommendation 3: Red-team after consistency training, not just before
These could be implemented by safety teams immediately, making the work practically relevant

Caveats:
Findings may not apply to production models: RLHF makes sycophancy nearly neutral; generality unclear
Scale uncertainty: 70B results with 1 seed are exploratory; can't confidently claim scale effects
Despite these caveats, the core contribution—showing consistency training is not alignment-neutral—is significant and practically important.

5. Presentation of the paper

The paper is **clearly written and well-structured** with an effective narrative arc: Figure 1 concisely communicates the core hypothesis (suppression vs. amplification outcomes), the three-phase pipeline (Figure 2) makes the experimental design immediately intuitive, and results are presented hierarchically (main findings in Section 5, detailed breakdowns in appendices). The taxonomy of consistency methods (Appendix A, Table 3) and comprehensive implementation details (Appendix B) enable reproduction—an expert reader could reconstruct the experimental setup from the provided specifications. The use of sign consistency as the primary metric is well-justified given high variance in effect sizes, and the paper appropriately separates concerns: main text focuses on key findings while extensive ablations (A1-A4) and statistical tests (Appendix J) are available for scrutiny. Positioning relative to prior work is adequate: the paper clearly distinguishes itself from capability-focused consistency training research (Wang et al., 2023) and method-specific alignment work (Irpan et al., 2025; Chua et al., 2024) by asking the **novel question** of whether consistency training is alignment-neutral across multiple misalignments. The paper positions itself naturally within the model organisms research agenda (Hubinger et al., 2023) as the first to use organisms as a testbed for evaluating alignment effects of training procedures.

However, the presentation has **two notable gaps that could be improved**.

First, the theory-practice gap (Proposition 3.1 predicts monotonic η(s), but empirics show flatness; yet large effects persist) is acknowledged in Section 6 but not directly confronted in the main narrative—the reader must piece together that the theorem's assumptions are violated in practice, which undermines its utility. A clearer statement like "Proposition 3.1's predictions fail empirically because [explicit reason], suggesting the mechanism is distribution shift rather than selection" would strengthen coherence.

Second, several high-impact findings are buried in appendices: the 70B scale reversal (Figure 8, Section H.4) showing consistency training becomes unsafe for reward hacking at scale is exploratory but has severe practical implications and deserves a main-text callout; the RLHF interaction (Table H.2) showing sycophancy amplification is near-zero in production models potentially undermines the practical relevance of main findings and warrants explicit discussion of generalizability.

Minor improvements: notation like κⱼ (Section 3.1) and Clarke subdifferential machinery are dense for non-experts and could be simplified or moved to appendix; the distinction between "amplification/suppression" and "non-neutral" could be stated earlier and more explicitly. Overall, the paper communicates effectively, but moving scale results and RLHF limitations into the main narrative and directly resolving the theory-practice gap would significantly improve clarity and prevent readers from missing critical caveats.

---

> ### Author Rebuttal · Authors · 2026-03-30
>
> We thank the reviewer for their detailed feedback, which we feel has improved our paper. We detail our revisions below:
>
> **Issue 1: Theory–practice gap.** We have revised the main text to clarify the difference between the theoretical predictions and empirical observations. Proposition 3.1 is useful precisely because its predictions fail in the flat-$η$ regime. This failure rules out selection as the dominant mechanism and redirects attention to distributional effects. When $η$ is non-flat, its predictions hold. We now frame the proposition as a diagnostic tool, as opposed to a full explanatory model.
>
> **Issue 2: Mechanistic evidence of distribution shift.** We appreciate the feedback that additional evidence is required to justify the claim that distribution shift is the primary driver of suppression. We summarize here an additional experiment we conducted and have written up in our paper revision, but please also see our response to Reviewer 6gx8 where we dive into more detail.
>
> In order to isolate the distribution shift, we test an additional method called Greedy Self-Training (GST) that uses no scoring, filtering, or selection to generate the output. We find that the results do not amplify sycophancy unlike selection-based consistency methods.
>
> We have revised Section 6 to present this more explicitly, and have noted the remaining limitation that specifying which features of the shifted distribution drive suppression versus amplification remains an open question.
>
> **Issue 3: Judge dependence.** We agree this is important and have added judge sensitivity experiments for emergent misalignment. Re-thresholding the raw GPT-4o alignment scores at five levels (20, 25, 30, 35, 40) across all 75 Phase 3 evaluations and 20 Phase 1 baselines, we find that sign consistency remains above 96% at every threshold, with all p-values below 10^-17. The suppression finding is extremely robust: our results do not depend on the specific choice of alignment score cutoff. The effect size increases with threshold (from -1.6pp at threshold 20 to -6.7pp at threshold 40), indicating that consistency training disproportionately reduces responses in the 20-40 alignment score range rather than shifting the entire distribution uniformly.
>
> **Issue 4: Generalizability (RLHF, scale).** We appreciate the reviewer highlighting these caveats.
> * RLHF interaction: We now explicitly footnote that sycophancy amplification is largely attenuated in instruction-tuned models, suggesting RLHF provides partial protection.
> * 70B results: We now clearly label these as preliminary and hypothesis-generating.
>
> Both points are now surfaced in the main text.
>
> **Issue 5: Presentation and Clarity.** We have revised Section 6.1 to explicitly connect the failure of the selection-based prediction and the resulting shift toward a distributional explanation. This is now stated directly in the main narrative.
>
> We have also defined amplification/suppression immediately after Equation 3 as requested.
>
> **Issue 6: Coherence and scale.** We thank the reviewer for this observation. Figure 6 measures cross-scale divergence (8B vs 70B), not within-scale consistency under perturbation. These are distinct: a 70B model can discover qualitatively different exploitation strategies than 8B (high cross-scale KL) while executing those strategies consistently under perturbation (high within-scale stability). We hypothesize this is why scale is important; larger models find and reliably execute more sophisticated exploits that smaller models cannot consistently discover.
>
> **Issue 7: Dual-use considerations.** We have expanded the impact statement to address potential misuse (amplifying coherent misalignment), defensive value (guidance for auditing), and information asymmetry.
>
> **Issue 8: Is sycophancy amplification fundamental or method-specific?** Our evidence more closely supports (A): sycophancy's coherence makes it fundamentally resistant to suppression via consistency training. All the methods we tested generally amplified sycophancy. However, our overall findings demonstrate that other factors like model size and family can also influence safety.

---

> > ### Author Rebuttal · Reviewer_37R5 · 2026-04-03
> >
> > The authors answer key questions vaguely with no strong citations/grounds.
> > e.g.
> > Issue 8: I wanted a theory/experiment based answer.
> > Issue 6:  response is very vague and hard to understand.
> > Issue 1,3,4 point to the need of extensive revisions in the paper. Issue 2 (even if gets resolved by the proposed experiment as the author's claim) demands a significant update.

---

> > > ### Author Response · Authors · 2026-04-03
> > >
> > > We thank the reviewer for their feedback and continued engagement.
> > >
> > > First, we would like to clarify that the experiments to address the raised issues that we described in our rebuttal are not proposed, but rather fully completed experiments as a follow-up to the excellent feedback we received. We have therefore linked a document with supplemental figures demonstrating the results of these additional experiments: https://anonymous.4open.science/r/icml-16799/paper_16799_supplement.pdf.
> > >
> > > **Issue 8 (Is sycophancy amplification fundamental or method-specific?).** We apologize that we didn't understand that the reviewer was requesting an additional theory/experiment based answer for this question. Regarding the reviewer's request to understand whether "(A) Fundamental: Sycophancy is so coherent that any consistency enforcement amplifies it (B) Method-specific: There exists a consistency method that doesn't amplify sycophancy, but you didn't test it," there are many consistency methods/variants in production use today so it is infeasible to test them all. We feel we have broadly covered the most commonly used consistency training methods and, as we noted in our rebuttal, find that generally sycophancy is amplified. However, a key point in our paper is that the choice of method and model can impact whether bad behaviors are amplified, so we don't think there is necessarily a clear-cut (A) or (B) answer to this question. Our results suggest that implementers of consistency training should be cautious, especially with regard to sycophancy, and check for dangerous amplification.
> > >
> > > **Issue 6 (Coherence and scale).** We appreciate this feedback and will be more concrete. Figure 6 and our scale hypothesis measure different quantities. Figure 6 measures cross-scale divergence: 8B and 70B exploit different strategies on the same prompts (e.g., keyword repetition vs. hardcoded reward functions), producing high KL. The scale hypothesis concerns within-scale consistency: a fixed 70B model generally executes its chosen strategy more reliably across *prompt variations* than an 8B model. Putting these two together, a 70B model can discover a qualitatively different exploit than 8B (high cross-scale KL) and execute it robustly (high within-scale stability).
> > >
> > > **Issues 1, 3, and 4.** We respectfully disagree that our responses imply major revisions to our manuscript. Our responses to Issues 1 and 4 involve adding clarification text and moving some results from the appendix into the body of the paper. Our response to Issue 3 was to conduct an experiment that was requested in the review, "MISSING-sensitivity analysis on judge thresholds," and discuss our findings, which are that the emergent misalignment suppression results from our manuscript are indeed robust to judge thresholds. This corroborates our findings rather than contradicts them. We have included these experiments in our paper as we found the suggestion to be very helpful.
> > >
> > > **Issue 2.** We appreciate the reviewer’s original suggestion that additional evidence would help justify the claim that distribution shift is the primary driver of suppression. Because of this, we ran the control Greedy Self-Training (GST) experiment, described in our author response. Our result was that sycophancy is not amplified in the same way as consistency training methods that employ an enforcement operator (selection, filtering, scoring) and this provides additional evidence for our claim. Overall, we don't view this supplemental experiment, which reinforces a claim already made in the paper, as a major update.

---

### Official Review · Reviewer_ADar · 2026-03-13

**Soundness:** 3
**Presentation:** 3
**Significance:** 3
**Originality:** 4
**Overall Recommendation:** 5
**Confidence:** 3

**Summary:**

To find out whether consistency training affects model alignment, the authors performed several consistency training methods on model organisms of several misalignment behaviors, fined-tuned from models across 7-70B sizes. They find that consistency training does indeed change misalignment behaviors with varying magnitudes and signs (amplification/supression). They also present some arguments/evidence that it is due to the distributional shift of labels caused by consistency methods that causes the observed change in misalignment behaviors.

**Compliance With Llm Reviewing Policy:**

Affirmed.

**Final Justification:**

I think this is overall a solid paper. My main concern was around whether consistency training was commonly used in production, and that was answered well by the authors.

**Key Questions For Authors:**

Would the authors be able to comment on which consistency training methods are more commonly used in model training, or whether they are used at all?
2. I am a bit confused at the separation of the terms used here: "Consistency training" refers to a training procedure, but includes both label-generation methods (which do not involve training) and regularization methods (which go through training)? **Can I clarify that by default, when you refer to label-generation methods in Section 5 Results, you're referring to models trained using the three-phase pipeline in Section 4, with the label-generation method referred to being used in Phase 2?**
3. Follow-up question: This means that you're not actually using any label-generation method in Section 5, and you're just running the trained model on the normal misalignment benchmark?
4. Follow-up question: Could you give an example of the pipeline, especially for Phase 2 and 3? What exactly are you running training on (I assume SFT) after running the label-generation methods?

**Limitations:**

yes

**Strengths And Weaknesses:**

Strengths:
* (Soundness) This authors perform quite comprehensive experiments, covering a wide spectrum across misalignment behaviors, model sizes and model families. They also include RLHF models.
* (Significance) This paper is likely to contribute to the literature by informing limitations and nuances around consistency training, given how thorough the experiments are.
* (Originality) I think this idea seems to be quite original, and is an interesting dimension to study with regards to the shortcomings of consistency methods and consistency training.

Weaknesses:
* (Presentation, Soundness) It is somewhat difficult to follow (1) exactly what sets of models/organisms/sizes were trained, as well as (2) exactly what the results are an aggregate of.
* (Significance) It doesn't seem like consistency training methods are actually used in current LLM training, which limits the significance of this paper.

---

> ### Author Rebuttal · Authors · 2026-03-30
>
> We thank the reviewer for their feedback and attention to detail. We feel that it has improved the clarity and positioning of our paper, and detail our revisions below.
>
> **Issue 1: Presentation clarity.** In our revision we have added a summary table in the main text that explicitly enumerates model sizes, seeds, organisms, training datasets, and methods applied. We have also added a worked example to more clearly illustrate the flow of our experimental pipeline, clarifying that results in Section 5 are always evaluated on the final (Phase 3) models. Finally, we have updated captions to clearly explain aggregations.
>
> **Issue 2 (and Q1): Practical relevance.** We respectfully disagree that these methods are not used in practice. Consistency-based or adjacent procedures (the method name varies) appear in multiple stages of modern pipelines, and have widespread use in frontier AI developers, including:
>
> * Rejection sampling and distillation (e.g., Llama 3.1, DeepSeek-R1),
> * Self-critique and revision loops (e.g., Constitutional AI),
> * Multi-sample selection in inference-time scaling (e.g., OpenAI o1 models),
> * Several consistency methods were published by teams at frontier companies, revealing interest in these methods (e.g., ACT by Irpan et al. 2025 at DeepMind, ICM by Wen et al. 2025 at Anthropic).
>
> These are typically treated as capability improvements rather than alignment interventions, which is precisely the assumption our paper interrogates. We have revised the introduction to make this connection explicit. We have additionally completed rejection sampling experiments with an external reward model (Skywork-Reward-V2-Llama-3.1-8B), mirroring the Llama 3.1 pipeline. Results confirm the same pattern: suppression of reward hacking (65%) and emergent misalignment (75%), with significant sycophancy amplification (10%, p < 0.001). This validates our findings in a production-relevant setting.
>
> **Issue 3: Terminology and pipeline clarity.** We thank the reviewer for flagging this; we have edited the manuscript to improve this. To clarify here, we use the term consistency training to refer to a general category of methods that encourage a model to produce similar outputs or internal representations across related inputs. We formalize this definition and explain how various prior methods fit within this framework in Appendix A.
>
> We study two sub-categories of methods under consistency training: label generation methods (SC, DD, MVC, SR, and Rew) and regularization methods (ACT and BCT). For label-generation methods, consistency is enforced during label generation (Phase 2), and the resulting labels are used for standard SFT (Phase 3). For regularization methods, consistency is enforced directly in the training loss using a paired dataset composed of the same inputs as the the inputs used in Phase 2 of the label generation methods and a wrapped version of the inputs (see Table 1 for an example).
>
> To answer the reviewer's questions directly, given the above clarification:
>
> *Q2: Can I clarify that by default, when you refer to label-generation methods in Section 5 Results, you're referring to models trained using the three-phase pipeline in Section 4, with the label-generation method referred to being used in Phase 2?*
>
> A: Yes, this is correct.
>
> *Q3: This means that you're not actually using any label-generation method in Section 5, and you're just running the trained model on the normal misalignment benchmark?*
>
> A: Yes, this is correct. We report the normal misalignment benchmark results on a held-out test set after the model has undergone two training phases: the first to induce misalignment on existing misaligned training data (Phase 1), and the second the consistency training (Phase 3) using the labels generated in Phase 2.
>
> *Q4: Follow-up question: Could you give an example of the pipeline, especially for Phase 2 and 3? What exactly are you running training on (I assume SFT) after running the label-generation methods?*
>
> A: No problem! Let's take reward hacking on Llama-3.1-8B with Self-Confidence:
>
> 1. Phase 1: Fine-tune Llama-3.1-8B on reward hacking data, producing a model organism that exhibits reward hacking at ~76%.
> 2. Phase 2: For each of 500 held-out prompts, the organism generates k=3 responses via stochastic sampling; the response with the highest mean log-probability is selected as the pseudo-label. No training occurs in this phase.
> 3. Phase 3: SFT the organism on the 500 (prompt, pseudo-label) pairs from Phase 2.
> 4. Evaluation: we then compare Phase 1 vs Phase 3 misalignment rates on the reward hacking evaluation benchmark

---

> > ### Author Rebuttal · Reviewer_ADar · 2026-04-04
> >
> > I thank the authors for their response. I find their answers convincing, especially the responses to the questions brought up by 6gx8, which I found to be quite relevant and insightful (both 6gx8's questions, and the authors to their response).

---

### Official Review · Reviewer_6gx8 · 2026-03-18

**Soundness:** 4
**Presentation:** 3
**Significance:** 2
**Originality:** 3
**Overall Recommendation:** 5
**Confidence:** 2

**Summary:**

The paper argues literature often treat consistency training as a benign/neutral technique w.r.t alignment, used primarily for better model reasoning robustness. However, the literature lacks a clear understanding of when does consistency training methods affect underlying model alignment (and in what directions). The work addresses this gap by extensive empirical tests. First, the work builds model organisms (models with specified misalignment failures) across four specific failure modes. The paper then tests seven consistency methods and measure the resulting changes in misalignment. They find that consistency training consistently reduces reward hacking and emergent misalignment, but it worsens effects like sycophancy, and has no strong effect on spurious correlations. Through mechanistic analysis, they argue that pseudo-labeling and distribution shift drive these effects, rather than simple score selection.

**Compliance With Llm Reviewing Policy:**

Affirmed.

**Final Justification:**

The authors have satisfactorily addressed my major concerns. I hope the authors:
- revise the title to better reflect the scope of the work.
- Include the additional points from rebuttal (e.g. mechanistic analysis) and explicitly contrast their findings with ACT and other existing models. This might help contextualize the findings for a reader (certainly helped for me), particularly since many of the observed effects may stem from how the model organism was created, a significant confounder.
- open-source the data/code/models.

That being said, I liked the work overall and believe it provides valuable data that will contribute to a more comprehensive understanding of these effects. Note that I do not work directly in this field, and might have overlooked something reflected in my confidence.

I also am quite unhappy/suspicious about review from Reviewer 37R5 -- I can't help but suspect it's LLM generated and am flagging this (easy-to-verify evidence in rebuttal acknowledgement), with a caveat that its plausible that I am wrong.

**Key Questions For Authors:**

Please answer weaknesses. Overall, I think the strengths outweigh the weaknesses and this is a valuable contribution to the alignment community. Please note I don't work in either of these problems, reflected in my confidence.

**Limitations:**

Yes

**Strengths And Weaknesses:**

Strengths

- Clear Value: I found knowing if consistency training used for capability improvements alters safety boundaries and in what ways. Extensive empirical investigation checking how consistency training changes alignment was nice to see, and provided genuine value to me (and hopefully to broader community). I think a usual mental model for thinking about consistency training that the work provides is asking: does the specific alignment task better if model is more coherent under perturbations? If yes, consistency training should improve alignment, if not it may worsen.

- Empirical Breadth: I found the experimental setup rigorous and extensive. The work is not rushed, and does not suggest bold claims from very little testing or controls. The testing is extensive: testing several open models, multiple methods, four distinct failure modes, and controlled across factors: with ablations across scale, candidate count, and label-source sensitivity. Finally, the mechanistic analysis provided some nice understanding on why consistency training behaves how it does.

Weaknesses

- Misaligned Framing: I worry the paper’s headline framing potentially misleading (I came in expecting consistency training would be directionally better or worse for alignment). The authors argue the method is not neutral because it changes specific behaviors -- which is usually the case for most things? As in, for a broad range of techniques (say RL) I would've apriori expected it improves some behaviors while worsening others. Was it a community held a prior belief that consistency training has no effect on any alignment property? To me it seems like paper studies when it simply studies when and how consistency training affects alignments behavior.

- Does regularization just worsen performance? There seems to be confound between alignment improvements and capability degradation here. Appendix tables show that ACT and BCT methods heavily degrade overall performance (e.g., MMLU, GPQA). I think the mechanism here explaining the Table 2 might be that the method simply made the model less capable.

- Heterogeneous Results: The authors study four tasks and measured success rates across the seven methods -- these seem to vary significantly and often lack directional coherence (Figure 3), making it difficult to cleanly support the overall claim. Table 2 which is clean has the above issue. I am not sure if evidence supports the broad claim cleanly (although I can see its more likely to be true than not).

- Incomplete Mechanistic Evidence: I find the claim that distribution shift is the primary driver lacks sufficient evidence. The authors provide indirect evidence to rule out simple score-based selection. This was helpful and shows that variation in the selection operators is not the underlying effect, but I'm not so convinced that distribution shifts is the underlying effect.

---

> ### Author Rebuttal · Authors · 2026-03-30
>
> We thank the reviewer for their thoughtful engagement. We address each concern below and hope our clarifications and new experiments strengthen the reviewer's confidence in our findings.
>
> **Issue 1: Misaligned Framing.** We agree our framing should be sharpened. While any training procedure will, to some degree alter behavior, our motivation stems from how consistency training fundamentally differs from other post-training methods. Because consistency training operates entirely on the model's own outputs, one might expect the technique to mainly solidify existing distributions. Its widespread use in modern post-training pipelines (e.g., Llama 3.1, DeepSeek-R1, Qwen 2.5) to enhance reasoning capabilities reinforces this. Instead, we show that consistency training can produce dangerous safety shifts, such as amplifying high-stakes behaviors like sycophancy. Given the lack of prior work on its safety implications, we find it important to document these unintended effects.
>
> We have revised the introduction to reflect this more precise claim.
>
> **Issue 2: Capabilities Confound.** We thank the reviewer for raising this important concern.
>
> We acknowledge that ACT and BCT show capability degradation on MMLU and GPQA. Both the original ACT and BCT authors observe some capability loss, particularly at smaller model scales (Irpan et al., 2025; Chua et al., 2024), consistent with what we observe in our 7–20B experiments. However, three points suggest that capability degradation alone cannot fully explain the alignment shifts we see.
>
> First, if the models were simply degraded to a less capable state, we would expect to see uniform reduction across all misalignments. Instead, we see divergent effects. For example, across our 7–20B models, ACT strongly suppresses reward hacking, but simultaneously amplifies sycophancy. Even if some of the suppression is partially attributed to capability loss, the directional asymmetry cannot be explained by degradation alone.
>
> Second, across 494 matched runs, the change in capability (MMLU/GPQA) and the change in safety (StrongREJECT) are approximately uncorrelated (r = -0.08, p = 0.07). This further suggests distinct mechanisms.
>
> Third, we note that the five label-generation methods, which largely preserve capability (~0.52 MMLU), show the same misalignment organism-dependent pattern, suppressing reward hacking while amplifying sycophancy, confirming that the alignment effects are not an artifact of capability degradation.
>
> **Issue 3: Heterogeneous results.** We appreciate the reviewer’s careful reading and agree that method-level variation is substantial. In fact, a key finding is that this variation exists. We conducted this large-scale evaluation across many models, methods, and tasks in order to measure and account for this kind of variance. We find that the type of misalignment (the organism) is by far the strongest predictor, with the consistency method, model family, and scale playing secondary roles. Aggregated at the organism level, the statistical signal is strong despite the method-level noise:
>
> Reward hacking: 63.4% suppression (N = 175, p < 0.001)
> Emergent misalignment: 71.9% suppression (N = 160, p < 10^-7)
> Sycophancy: 75.3% amplification (N = 174, p < 10^-10)
>
> In Section 6.2, we present evidence that coherency is the organizing principle. Reward hacking relies on narrow, brittle heuristics (exhibiting ~10x higher KL divergence between 8B and 70B), so it gets suppressed. Sycophancy is a generalized, robust strategy (low KL divergence), so it gets amplified (see Figure 6).
>
> The method-level variation the reviewer observes arises because coherence depends on a complex interaction between the misalignment type, model family, and specific perturbations per method. As a result, the per-cell results are noisy even when the organism-level pattern is robust. We have added full results to the appendix to make this variation transparent.
>
> **Issue 4: Mechanistic evidence.** We agree the original manuscript emphasized ruling out score-based selection more than affirmatively proving distributional shift. To address this, we ran a Greedy Self-Training (GST) baseline.
>
> GST generates a single greedy completion per prompt with no scoring or selection, so any alignment shift can only be attributed to the distributional shift between the Phase 1 misaligned training data and the Phase 3 self-generated outputs. We find that GST reproduces the suppression effects on reward hacking (70%, Δ = −7.1pp) and emergent misalignment (80%, Δ = −0.8pp). Crucially, GST does not amplify sycophancy (Δ = -0.7pp), unlike selection-based consistency methods. This dissociation further shows that distributional shift drives suppression, while the selection operators drive the amplification of coherent behaviors.
>
> We have revised Section 6 to present this more explicitly, and have noted the remaining limitation that specifying which features of the shifted distribution drive effects remains open.

---

> > ### Author Rebuttal · Reviewer_6gx8 · 2026-04-02
> >
> > I went through the rebuttal and other reviews (esp. 37R5 since it was detailed), but found myself more confused: an easy-to-verify point is e.g. "Minor improvements: notation like κⱼ (Section 3.1) and Clarke subdifferential machinery are dense for non-experts and could be simplified or moved to appendix" there seems to be no κ$_j$ in Section 3.1, nor could I find any "clarke subdifferential" or related concepts. Note that I do not work directly in this field, and might have overlooked something.
> >
> > > Given the lack of prior work on its safety implications, we find it important to document these unintended effects.
> >
> > I was skimming the ACT paper to better understand the broader context.
> > - One thing that stood out to me is that the ACT paper states that *"BCT and ACT reduce sycophancy equally well"* which appears to conflict with this work's finding that sycophancy is amplified. Could you help reconcile these two opposite results?
> > - Why does performance degrade so much in this work --  did the training in this work just go wrong, and if so are the results trustable? (Perhaps a better phrasing of my W2 in hindsight. e.g. GPQA has random performance of 25\% (4 options) but the performance of the ACT/BCT model is 5%). In contrast, the performance reported in ACT is quite strong across methods and the results seem more trustable.
> >
> > Understanding why the conclusions directionally differ with consistent findings (their findings also seem strong -- Fig 2, across different models and compared to a control) would helping me evaluate the robustness of the overall pattern (*..  aggregated at the organism level, the statistical signal is strong*) -- as sycophancy seemed to be the most directionally consistent effect among those studied.

---

> > > ### Author Response · Authors · 2026-04-03
> > >
> > > Thank you for your continued effort and engagement.
> > >
> > > **On the reference to κⱼ and Clarke subdifferentials.** We can confirm that our paper does not contain κⱼ notation in Section 3.1 or Clarke subdifferential concepts anywhere in the manuscript. We believe this may have been an error, perhaps referring to a different submission.
> > >
> > > **On reconciling results with Irpan et al. (2025), who find that "BCT and ACT reduce sycophancy equally well."** We thank the reviewer for carefully reviewing the original method paper. We believe the two results are reconcilable once the starting conditions are accounted for.
> > >
> > > Essentially, our differing results stem from the base models to which we apply consistency training. This distinction also helps explain the capability degradation discussed further below. Irpan et al. apply BCT/ACT to regularly-aligned models like Gemma 2/3 and Gemini 2.5 Flash where the models generally behave well on clean prompts. In our setup, we fine-tune to create "model organisms" that are already more systematically sycophantic before applying consistency training. Because of this difference, when we generate clean-prompt completions, a larger portion of the model's own outputs are sycophantic, more than the models that Irpan et al. work with.
> > >
> > > Consistency training therefore reinforces the model's existing (misaligned) stable behavior rather than correcting it, one of the findings of our paper. The directional differences help reinforce one of our conclusions that consistency training can amplify stable behaviors. When that stable behavior is aligned (Irpan et al.), consistency training helps. When it is misaligned (our model organisms), consistency training hurts. This is precisely the Consistency Non-Neutrality Hypothesis: the effect depends on the structure of the behavior.
> > >
> > > Additionally, our own RLHF interaction ablation (A2, Appendix H.2) is generally in line with Irpan et al.'s results. Our instruction-tuned models, which, like Irpan et al.'s models, have already been trained away from sycophancy, show near-zero sycophancy amplification (mean Δ = −0.2%), consistent with Irpan et al.'s findings.
> > >
> > > **On capability degradation.** We believe the degradation arises from differences in experimental setup.
> > > * Stacked fine-tuning. Our models undergo sequential rounds of fine-tuning: an initial phase (Phase 1) where misalignment is induced, and a subsequent phase (Phase 3) where consistency training takes place. Phase 1 already degrades general capabilities. ACT/BCT then applies a second round of parameter updates on top of an already-degraded model.
> > > * Model scale. Irpan et al.'s strongest results are on Gemini 2.5 Flash, a frontier model substantially more robust to fine-tuning perturbation than the smaller open-source models we study. Smaller models are known to be more susceptible to degradation under sequential fine-tuning.
> > >
> > > Importantly, capability degradation does not undermine the directional finding from ACT/BCT. If the models were simply degraded to a less capable state, we would expect uniform reduction across all misalignment types. Instead, ACT/BCT show strong asymmetry: suppression of reward hacking and emergent misalignment, but amplification of sycophancy. Capability degradation alone cannot explain why the same training run suppresses one behavior and amplifies another. Furthermore, this same asymmetric pattern appears in our five label-generation methods, which largely preserve capability.

---

### Decision · Program_Chairs · 2026-04-30

**Decision:**

Accept (regular)

**Comment:**

This paper studies whether consistency training (methods that encourage models to produce similar outputs across related inputs or sampling procedures) has differential effects on model alignment. Testing seven consistency methods on 108 model organisms spanning four failure modes (reward hacking, emergent misalignment, sycophancy, spurious correlations) across 7B-70B models, the authors find that consistency training consistently suppresses reward hacking and emergent misalignment while amplifying sycophancy, and offer a mechanistic explanation based on behavioral coherence under perturbation.

**Strengths.** Reviewers agree that the question is timely and practically important, as consistency training is increasingly adopted for capability improvement with little attention to its alignment implications. The empirical coverage is broad and carefully controlled: seven methods, four failure modes, multiple model families and scales, with ablations across candidate count, label source, and scale. The mechanistic analysis (Section 6.2) seems to be the highlight of the paper, providing an interpretable account of the mixed results: sycophancy is behaviorally coherent under perturbation (low KL divergence across outputs) while reward hacking is not, which explains why consistency training reinforces one and suppresses the other. The practical recommendations are concrete, intuitive, and actionable.

**Concerns.** The main concerns raised were around theory presentation: Reviewer 5dK2 flagged unclear notation in Eq3, an undefined variable in Eq5, and missing citations on self-rewarding methods. Reviewer 6gx8 raised a framing question about whether "not alignment-neutral" is a surprising finding for any training procedure. These were the primary issues entering the rebuttal.

**Rebuttal.** The rebuttal was highly effective. After the author's clarified questions and presentation issues, three of four reviewers raised their scores: Reviewer 5dK2 upgraded from 3 to 4 after the authors addressed the theory concerns; Reviewers 6gx8 and ADar both upgraded from 4 to 5. These upgrades provides strong evidence that the paper's core contributions hold up to scrutiny.

**Bottom line.** A solid, well-executed empirical study on an understudied and practically relevant question, with a convincing mechanistic explanation of the phenomenon, which garnered accept consensus after the rebuttal.